# Modeling and Control of Supercritical and Ultra-Supercritical Power Plants: A Review



**Omar Mohamed** [1,*], **Ashraf Khalil** [2] **and Jihong Wang** [3]

1   King Abdullah II School of Engineering, Princess Sumaya University for Technology, Amman 11941, Jordan
2   Electrical and Electronic Engineering Department, Universiti Teknologi Brunei, Jalan Tungku Link,
    Gadong BE1410, Brunei Darussalam; ashraf.sulayman@utb.edu.bn
3   School of Engineering, University of Warwick, Coventry CV4 7AL, UK; jihong.wang@warwick.ac.uk
*   Correspondence: o.mohamed@psut.edu.jo

**Abstract:** This paper presents a critical review of the research conducted for modeling and controlling supercritical power plants. Thermal power plants are classified according to the boiler pressure to supercritical and subcritical. The modeling concepts and control strategies of supercritical generation units are far more complex than those of subcritical. On the other hand, supercritical generation technologies are more efficient and much cleaner than subcritical generation units. From a deep technical analysis of the literature, there is no review that is dedicated to models-based control of supercritical power plants and most previous reviews are found to be too general to modeling-based control of fossil fuelled energy sources. This review reports the advancements on modeling and control of supercritical and ultra-supercritical plants as cleaner generation technologies. The various published achievements for modeling supercritical and ultra-supercritical units have been reviewed. The control strategies that fulfill the practical load demand requirements while keeping optimum efficiencies are also reviewed. Finally, expected future directions are reported as recommendations to overcome future challenges. The paper can be used as a brief educational directory to the postgraduate students or future researchers in the field.

**Keywords:** supercritical power plants; ultra-supercritical power plants; modeling; identification; control

## 1. Introduction

Fossil-fired power stations produce undesirable environmental effect of $CO_2$ emissions which have to be reduced by an appropriate operation strategy and captured by a $CO_2$ capture technology. On the other hand, there is continuous growth in power demand because of increase in industrialization and population, which leads to a continuous need for fossil fuel generation technologies because of their flexible operation and large generation capacities. Although there is a huge number of research attempts for renewable energies and resources to support the fossil fuel technologies or eliminate their need, the renewable resources are generally not capable of covering the large proportion of the load request within a short period of time. This reason, in addition to other reasons related to the advantages of flexible generation units as compared with renewables resources, have introduced the righteous conclusion of indispensable power generation using fossil fuel units in recent time. However, modern thermal power stations must be more energy-efficient and less pollutant. Power plants with supercritical (SC) boilers have higher Rankine cycle because of high operating pressure and temperature (above critical points for water 22.12 MPa and 647.14 K pressure and temperature respectively [1]), which leads to greater energy efficiency, and hence lower fuel consumption and emissions. SC boilers exist in some coal-firing units, oil-firing units, and combined cycle gas turbine (CCGT) units. The later is differentiated by the regular SC boiler by other name, which is supercritical heat recovery steam

generator or HRSG [2]. There have been advanced recent studies conducted in the past decade to justify the replacement of the existing coal-fired power plants with these fuel-saving and cleaner units [3–6]. Supercritical generation units are one of the suitable technologies with improvements in environment and efficiency. Therefore, it is essential to establish international cooperative research capacity to suggest research solutions in response to the difficulties resulting from the choice supercritical power plant (SCPP) technology in electrical power generation and also to provide the training needed for future power engineers and operators. SC boilers are designed to be only of once-through structure because there is no state transition from water to steam phases in supercritical boilers so there is no need for drum to separate water/steam mixture and the no circulation as the case in subcritical units. The main advantages of fossil fuel SCPPs are summarized in the following two points:

- Because of their higher efficiency, they consume less fuel and consequently produce lower $CO_2$ and $NO_x$ emissions.
- The load demand following response is much faster than subcritical units.

However, despite the advantages of supercritical plants, they are widely known to be less reliable than drum subcritical plants in response to abnormal conditions or large disturbances such as partial load rejections [7]. Moreover, there have been some concerns to adopt this generation technology in some developed countries, mainly for the compliance with the grid-code (GC) [3,4]. The supercritical state of fluid means that there is not distinction between the liquid state and vapor state; this leads to unclear level of water in the waterwalls of SC boilers. Unlike the subcritical drum units, the water level is a variable that plays an important role in control system design. The SCPP area of research falls within a very sophisticated interdisciplinary area, which needs high collaborative research of power engineering, modern control engineering, mechanical engineering, and chemical engineering with varying focus proportions, depending on the objectives of the core study. Figure 1 demonstrates the multidisciplinary nature of the research area of SCPP and ultra-supercritical (USC) modeling and control. Therefore, intensive research must be established to conduct the whole process dynamic simulation of the various variables in SCPP, which are involved in the energy conversion process from fuel combustion to electricity generated through the infinite bus to simulate the process and normal and emergency conditions and how the simulated response could be improved. The published outdated and latest surveys are generalized for fossil fuel power plants whether they are subcritical or supercritical [8,9]. On the other hand, because of the above aforementioned differences between subcritical and supercritical fossil fuel plants, there has been an insistent need to revisit and organize the literature to present a distinctive and qualitative review for supercritical units in terms of simulated models and control.

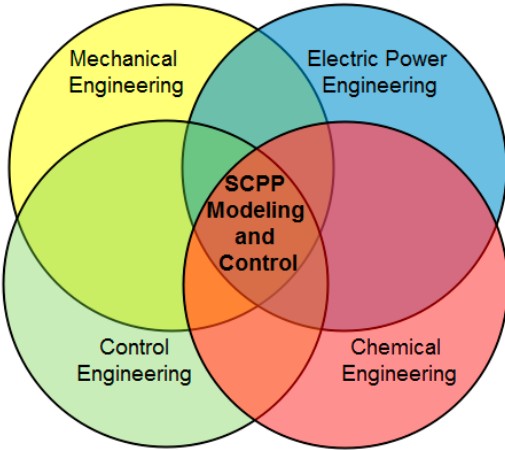

**Figure 1.** The interdisciplinary nature of supercritical power plant (SCPP) modeling and control.

The modeling significance is further explained. First in predicting some variables that are expensive and difficult to be accurately expected without the art of modeling, such as some intermediate variables of the SCPP. Thereby, SCPP modeling positively supports the hypothesis of adopting this technology as safe, reliable, and energy-efficient power generation system. The second important target for SCPPs modeling in particular is to facilitate the integration of $CO_2$ capture and storage for more friendly environment. The consequences of such important development will not be fully understood without realizing the modeling of the whole unit from fuel preparation to electricity generation. The third main significance for modeling is designing or upgrading the plant control. New operational objectives have been introduced to the system of the plant and thus the old control methods are obviously incompetent. Novel control strategies are supposed to capture wider power demands while maintaining the plant efficiency and economy at optimum level. The contribution of this paper is then to present a recent and critical review that justifies the adoption of SCPP technologies by reviewing the original research attempts for enhancement of SC and USC power plants' operation by modeling-based control and contextualize the various techniques as classifications to show the importance of this generation technology as separate research topic. Unlike most of the preview reviews that rather offer general reviews for modeling and control of fossil fuel units, which can be subcritical or supercritical. Moreover, this paper suggest new trends for future researchers, which can be developed to be salient PhD topics. Thereby, it provides an educational guide for future researchers to conduct original research in the field.

The rest of the paper is organized as follows: Section 2 presents an overview of supercritical power generation process; Section 3 reports the modeling approaches and control techniques in separate sections. Section 4 suggests some future trends in the field and Section 5 states concluding remarks about the comprehensive review.

## 2. Coal-Fired Supercritical Power Generation Process

In this section, the energy conversion process of SCPP is briefly described, assuming coal-fired SCPP schematic with the major plant subsystems as shown in Figure 2. The coal entering the mill is ground to a powder, which flows to the furnace where it is fired. The heat energy released from coal combustion is transferred by various ways of heat travelling, mostly by conduction and convection to the water inside the heat exchangers. The water is supplied by using the feedwater pump to enter the various heating stages in the SC boiler, the feedwater heater, the economizer (ECON), the waterwalls, and the superheater (SH). The supercritical or ultra-supercritical USC conditions occur in the waterwall and thereafter. The superheated supercritical steam flows to the steam turbine, in which the thermal energy is converted into mechanical energy. The efficiency is highly influenced by the enthalpy drop and the expansion in the turbine. The turbine can be designed to be combined.

Turbines to provide an appropriate steam production are designated as the high pressure (HP) turbine, the intermediate pressure (IP) in the figure, and even to low pressure (LP) turbine in practice. The steam exhausted from the HP turbine is fed back to the reheater (RH) to increase the heat content in steam before it is sent back to the IP and LP turbines. The induced draft fans push the flue gas to emanate from the stack. The rest of unit parts can be readily tracked by the figure. The next section reviews the modeling techniques and identification of the various plant components.

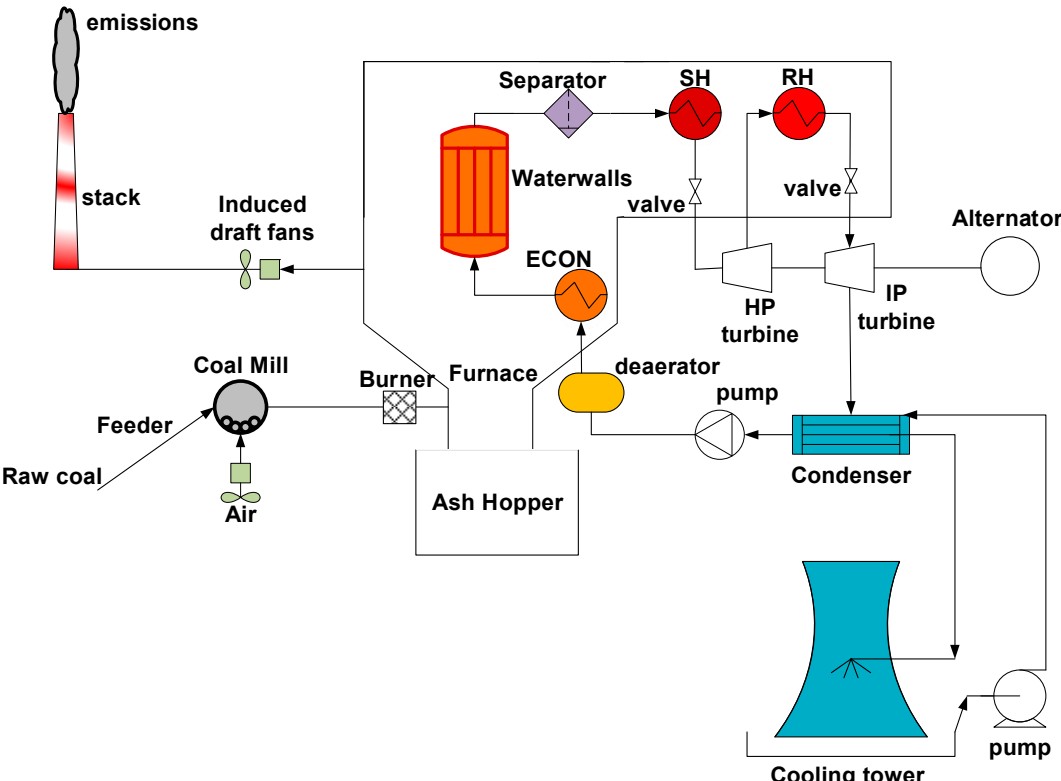

**Figure 2.** Coal-fired SCPP schematic view.

## 3. Supercritical Power Plant Modeling for Dynamic Performance Studies

The modeling techniques for SC and USC units fall within two main categories:

1. Models rooted from thermodynamic and engineering principles or simply physical models and mathematical models.
2. Empirical models or black-box modeling.

For proper classification of these methods the physical or thermodynamic modeling is first described. The plant once-through mode cannot be immediately brought by the plant after grid synchronization. The plant usually starts operating on this mode from 30% of the output power. The recirculation mode is needed after grid synchronization in order to attain appropriate temperature, pressure, and hence water level, then, the once-through model is safely adopted by closing the recirculation valve. The next section summarizes the fundamentals of typical modeling and reviews the models in the published research work.

### 3.1. Physical and Mathematical Modeling of SC and USC Power Plants

The concepts of modeling SCPP are based on mass and energy conservations of the control volume. The difference between the terms physical and mathematical modeling can be known through the adopted simplifying assumptions that are sometimes needed to convert the model from detailed physical model to simplified mathematical model in order to reduce the computational demands and facilitates computer implementation. In this review, some salient features of physical modeling SCPP are included. The fuel preparation system, the boiler, and the turbine are divided into some control volumes that represent the major components of the plant. The transferred energy concept holds the whole process. Mathematically, they can be described as [10]:

$$\frac{dE_{sys}}{dt} = \dot{E}_{in} - \dot{E}_{out} \tag{1}$$

The energy balance concept is established such that the net converted or energy equalizes the rate of change of the energy inside a system component, which represents the dynamical changes in the internal energy, kinetic, and potential energies occupied by the thermodynamic device. However, for steady-flow processes, the energy balance for the cycle is as follows:

$$\dot{E}_{in} = \dot{E}_{out} \tag{2}$$

and the mass balance principles is formulated:

$$\frac{dm_{sys}}{dt} = \dot{m}_{in} - \dot{m}_{out} \tag{3}$$

There have been methodical steps for developing models for fossil fuel units [11–54]. The dominant assumptions on the published research throughout the progress are: (a) the uniform properties of fluid; (b) the steady flow process; (c) excluding the flue gas path, and so on. For the boiler and the turbine, the technique is based on dividing the power plant into components. Each component or control volume is governed by differential equations that describe the dynamics of pressure and temperature or mass flows inside the control volume. Figure 3 uses the concept of control volume.

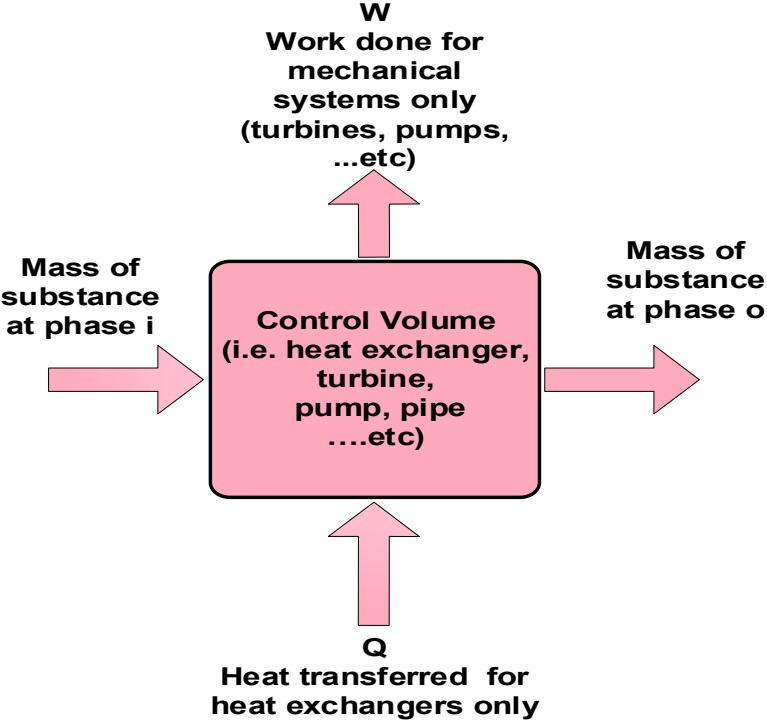

**Figure 3.** The control volume.

The dominant structure of the models has three-input three-output for multivariable control frame work. Physically speaking, it is insufficient to focus on load demand following while ignoring other safety restrictions. Once the turbine valve is opened to admit more thermal energy to the turbine, the pressure and temperature in the boiler are dropped, and thus there will be a need to increase the fuel flow and feedwater flow to continue the energy production, preserve the heat balance in the boiler, and keep the temperature and pressure on the desired values. The basic inputs/outputs scheme is shown in Figure 4.

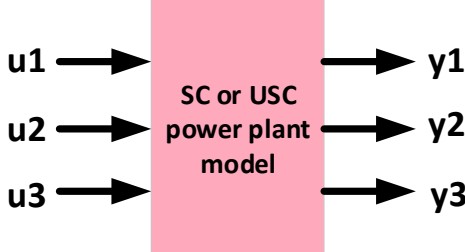

**Figure 4.** 3-input 3-output supercritical (SC) or ultra-supercritical (USC) model.

The input vector $U = [u_1 \quad u_2 \quad u_3]$ corresponds to the feedwater flow, the fuel flow, and turbine control valve position, respectively. Other indirect inputs can be included according to the control system objectives, but these are the ones that fulfill the basic operation requirements. The output vector $Y = [y_1 \quad y_2 \quad y_3]$ contains the generator's output power, boiler's temperature and pressure. The outlet variables of the boiler may be considered as the same as superheater pressure and temperature for simplicity in modeling. The most sophisticated part in the plant is the SC boiler so one should present some details about SC boiler modeling and its interactions with other components. It will be also useful to write the model equations in nonlinear state-space format. The equations of the heat exchangers inside the SC boiler are mathematically derived to get the pressure and temperature dynamical equations, which may lead to the following nonlinear state's format of equations' pairs for each heat exchanger in the SC boiler [3,37,38]:

$$\underbrace{\dot{x}_1 = \frac{\dot{Q}_{ec} + K_1 u_1 - K_2 \sqrt{x_1 - x_3}}{C_1} \quad \dot{x}_2 = K_3(a_1 u_1 - a_2 \sqrt{x_1 - x_3}) - K_4 \dot{x}_1}_{\text{Economizer}} \tag{4}$$

$$\underbrace{\dot{x}_3 = \frac{\dot{Q}_{ww} + K_5 \sqrt{x_1 - x_3} - K_6 \sqrt{x_3 - x_5}}{C_2} \quad \dot{x}_4 = K_3(a_3 \sqrt{x_1 - x_3} - a_4 \sqrt{x_3 - x_5}) - K_4 \dot{x}_3}_{\text{Waterwalls}} \tag{5}$$

$$\underbrace{\dot{x}_5 = \frac{\dot{Q}_{sh} + K_5 \sqrt{x_3 - x_5} - K_6 \frac{x_5}{\sqrt{x_6}}}{C_3} \quad \dot{x}_6 = K_7(a_5 \sqrt{x_3 - x_5} - a_6 \frac{x_5}{\sqrt{x_6}}) - K_8 \dot{x}_5}_{\text{Superheater}} \tag{6}$$

$$\underbrace{\dot{x}_7 = \frac{\dot{Q}_{rh} + K_9 \frac{x_5}{\sqrt{x_4}} - K_{10} \frac{x_7 u_3}{\sqrt{x_8}}}{C_4} \quad \dot{x}_8 = K_{11}(a_7 \frac{x_5}{\sqrt{x_4}} - a_8 \frac{x_5}{\sqrt{x_8}}) - K_{12} \dot{x}_5}_{\text{Reheater}} \tag{7}$$

where $K_1$ to $K_{12}$, $a_1$ to $a_8$, and $C_1$ to $C_4$ are unknown parameters to be identified. For every numbered pair of equations, there are two state derivatives that generally describe the sate variables in every heat exchanger in the SC or USC boiler. For example, the heat exchanger states $\dot{x}_1$ and $\dot{x}_2$ are the state derivatives of the pressure and temperature in the economizer, respectively, $\dot{Q}$ is the rate of change of the heat transfer of the relevant heat exchanger according to the subscript, for instance, $\dot{Q}_{ec}$ is the heat transfer rate to the economizer, and so on. The other pairs of equations are defined as shown above for state variables (i.e., temperature and pressure) of the waterwalls, the superheater, and reheater, respectively. The heat transfer rates can be related through transfer function to the input fuel flow $u_2$. The derivation of the aforementioned model step-by-step from the conventional conservation principles up to their final format can be found in previous work of the corresponding author [3,37,38],

with pressure and temperature symbols rather than state equations' format. The turbine and generator models are easier to develop and implement than the boiler model. However, there are direct mutual interactions that affect the turbine and generator unit that must be taken into account to have realistic simulations.

For instance, if the turbine speed is reduced, it is translated to be underfrequency in the generator that causes overflux in the generator magnetic circuit, which eventually overheats the electric machine above the normal. On the other hand, the underfrequency in the electric grid may be interpreted to be mechanical tenseness resonant in the turbine speed. The boiler and turbine units have also some mutual effects, such as that caused by opening the control valve of the turbine, which leads to pressure drop in the boiler. This situation demands an increase in feedwater flow and fuel flow rates in order to save the heat balance and pressure in the boiler.

Figure 5 demonstrates the interactions between these components, which shows that simply combining these components is sequential manner is not physically recognized.

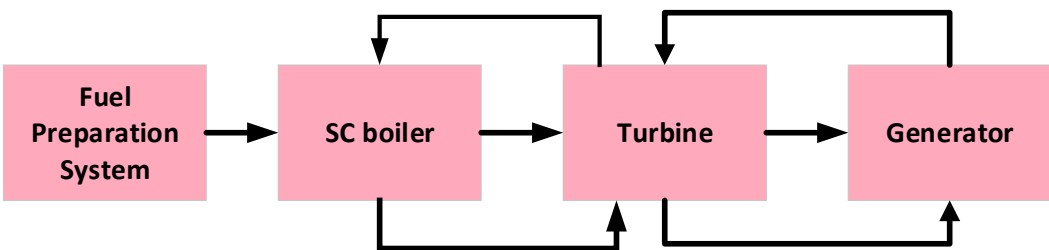

**Figure 5.** The boiler-turbine-generator physical interactions.

The various models' parameters can be determined from thermodynamic properties, which form the distributed-parameter models, or identified by multi-objective optimization technique to match the response of on-site data of the plant. Figure 6 represents the flow diagram for the systematic approach of physical modeling, which is applicable for SCPP and even other dynamical systems.

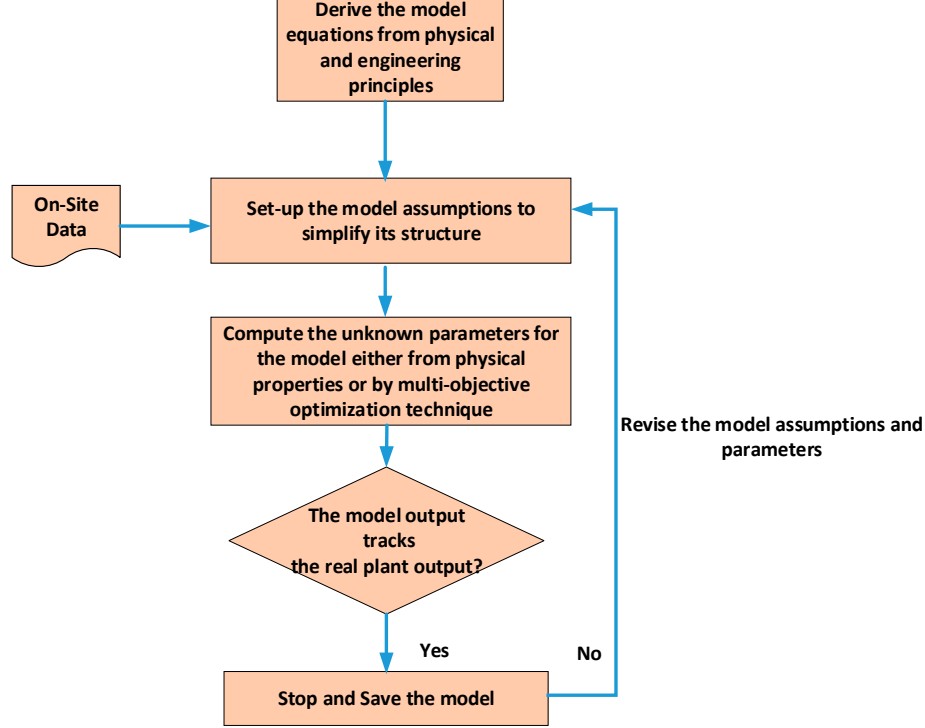

**Figure 6.** Physical and mathematical model parameters identification flow diagram.

The review for SCPP physical modeling is the following. It is intended to deliver the complete story in the literature and cover wide range of published research, new and old, for two main reasons: First, some old papers contain important justification of the fundamental assumptions and concepts of modeling and control, which help preliminary researchers in the field to have clearer picture of research progress. Second, the review covers the very new up to 2020 with detailed comments so it will not be vain to include old reference, especially when some of them are useful for both prominent and preliminary researchers in the field. Furthermore, the analysis should be quantified by powers as part of the assessment of the model or controller performance. For example, some models cover wider range than others so we will be able to know that by quantified power variations. The case is more important for controllers to know the capability of controller for load following. Research on the dynamics and controls of SCPP was initiated for the first time for simulating Eddystone I generation plant owned by Philadelphia Electrical Utility on 1958 and the that model was upgraded for simulating Bull-Run SCPP, in which many modifications and enhancements had been included [31]. The SCPP was formulated via linearization of the model that is based on physical fundamentals. However, there were no practical sets of data available for validation and that unit was not in full operating conditions at that time.

Laubli et al. [32,33] have depicted both practical experiments and time-based simulations to perform mathematical experiments for SCPP (gas and coal-fired with different specifications) and have proved the capability of these units in following the load demand rapidly and have validated the classical coordinated control performance for flexible operation of SCPP. No details have been given to describe the modeling derivations for the boiler and only computerized model with lumped parameters has been represented. However, the rotating elements, such as turbines, generators, and fans have been regarded to be single-flywheel storing energy of rotation whenever the speed is raised. Simulations have proved the capability of SCPP as flexible generation technology for base-load power and rapid demand following.

Yutaka Suzuki et al. [34] modelled a 450 MW oil-firing supercritical unit in order to enhance the existing design of control system of the practical SCPP. The SC boiler model has been based on several nonlinear partial differential equations, which formulate the mass, energy, and momentum conservations. The assumptions are refined to include some other details that have not been previously implicated. The model has been linearized and the unknown parameters are identified in such a way to fit the operation data records. The model structure has forward and backward coupling relationships between the various boiler components, which contribute to preserving high accuracy over a wide range of operation. The inputs are the fuel flow, water flow, and the steam fed back to the boiler from the turbine, whereas the outputs are the delivered steam flow rate, pressure, and temperature. Simulations have indicated the validity of the physics-based model.

W. Shinohara et al. [35] have shown a simple nonlinear third-order model from control point of view for supercritical boiler-turbine unit. The input is the heat proportional to the fuel flow input, and the output is the generated power. The simulation results are compared to agree with EPRI simulator results rather than practical data for a power drop from 510 MW to 450 MW with reasonable agreement between the two simulators. However, the steam pressure dynamics are assumed to be zero, which appear to be not consistent with other commonly adopted simplifying assumptions.

Toshio Inoue et al. [36] have developed a pressure node formula for power system frequency excursions simulation study and to justify the use of low order models. The development of each boiler section has been based on mass and energy conservation principles. The model is dedicated for power system frequency simulation studies. The model parameters are identified through comparison with another detailed model and the depicted results have been promising with the higher changes in the power normalized from 1.03 pu to 0.77 pu.

However, despite the outstanding effort that have been introduced in all aforementioned research articles for understanding the SCPP characteristics with the power grid that have lead to many practical advances in the field, there were no sophisticated technique offered for computing the unknown

parameters and all such parameters have been computed simply through direct comparison with real measurements, detailed model, or from thermodynamic properties. Henceforth, more accurate results seem to be possible, even with simplified models, especially with advent of the mathematical optimization and system identification sciences. In the past decade, more effort has been given to refine the parameters identification procedure in order to produce modern versions of SCPP models.

Mohamed et al. [3,37,38] have modelled a 600 MW SCPP using engineering and thermodynamic fundamentals with some simplifying assumption. The boiler model has been based on the boiler thermodynamic. The turbines and their interactions with the synchronous generator have been modelled via mass-spring system, in which the shaft behaves like spring connecting the masses of the turbines and the rotor of the synchronous generator. The generator has been modelled using power angle equations in addition to the EMF and power equations that are presented in q-d axis. Finally, the model with its unknown parameters has been represented in MATLAB/SIMULINK® computer environment. The unknown model parameters have been identified using evolutionary computation technique, mainly genetic algorithms (GA). The model accurately follows the dynamical data of the plant for the whole once-through mode as the power changed from 210 MW up to 600 MW. Moreover, apart from verifications with on-site data dynamic responses, the model has been compared with Thermolib for static performance of the model. Thermolib is a generic computer tool, expanded from MATLAB®, that simulates thermodynamic systems developed. However, it can be noticed that Thermolib is not capable of handling the properties of solid states of materials. Therefore, the coal preparation system or milling process has been considered by extending the data file of Thermolib with the gasified coal properties in order to capture the whole process. Again, the simplified SIMULINK model has shown strict agreement with Thermolib as well as the plant real measurements.

M. Draganescu [4] has presented a detailed model for the same 600 MW SC power plant. The model is detailed with more process components, such as the flow resistance model, the turbine valve, the fan and pump, deaerator, and so on. However, the fluid network philosophy is based on mass and energy balance concepts. The model is derived and implemented via FORTRAN 95 programming language and its parameters have been optimized and refined several times by research group at Tsinghua University. However, the research objective in [4] has been to study the compliance of SCPP with Great Britain (GB) Grid-Code and the main contribution is to introduce the dynamic matrix control (DMC) for improving the plant responses, which lead to verification of the ability of SCPP to fulfill the grid-code requirements.

Guo [5] has modelled the post combustion carbon capture process (CCP) and has integrated it with the SCPP models in [3,4,37,38]. Moreover, some fixed parameters are dynamically presented in the refined version of the model to implicate the effect of other thermodynamic variables, such as the fluid enthalpy. The post-combustion carbon capture process has been modelled by the same first principles, which involves four main components: the absorber, the regenerator, the reboiler, and the main heat exchanger. Thermodynamic properties have the major impact on the parameters calculations by look-up table linked to the equations used to describe the process. Finally, the impact of the CCP on the whole plant process has been studied through wide range dynamic simulations. This research rather introduces another important purpose of SCPP modeling, which is related to the carbon capture technology advancement.

J. Z. Liu et al. [39] have developed a dynamical model for 1000 MW ultra-supercritical power plant from physical laws and some simplifying assumptions and have built the derived model in MATLAB®. For static and dynamic simulation the model has shown reasonable agreement. The parameters of the dynamical model have been identified from closed-loop data using an intelligent technique, which is genetic algorithms technique. The final format of the model has been represented as state-space format. The plant output dynamical changes have been relatively small as the power output is compared with the model simulations with load variations from around 612 MW to 590 MW in the validation phase, whereas the static computations cover the data variations from 505 MW to 1000 MW.

Because the model have matched the on-site data measurements, the model has become feasible for control design purposes.

Alobaid et al. [40] have investigated the startup and load following processes for a supercritical heat recovery steam generator (SC HRSG) with its associated controls in order to attain high level of accuracy for wide range multi-processes, which are the startup and load following processes. The SC HRSG is well-known as a subsystem of combined cycle power plants and the study has been conducted for efficiency analysis and improvements purposes. The developed model is an extended version of another model developed for subcritical power plants. Advanced Process Simulation Software (APROS®) has been utilized to conduct the process simulation in static and dynamic conditions, which lead to comparative analysis for the assessment of efficiency and steaming conditions between subcritical and supercritical units.

Deng et al. [41] have introduced a new dynamic simulator, based on physical modeling, for start-up process dynamic simulations using (APROS®) commercial software package, which is advanced process simulation software. The contribution of the work is realized by simulating the start-up process of 600 MW SCPP by homogenous and two phase modeling from mass, momentum, and energy balance laws. The start-up process covers the process from fringing the boiler up to 30% or loading. Two different start-up processes have been considered, which are long-term start-up and start-up after 72 h shutdown. The simulator has shown reasonable matching between its outputs and the design values and curves in static and dynamic simulations, respectively. The model coefficients are calculated using thermodynamic formulas to be implicated in the pressure equations of the various boiler components. This model can be used in optimizing the future start-up processes or operator training simulation purposes.

Sarda et al. [42] have widely modeled the SCPP in Aspen Plus Dynamics® simulation environment (APD®). The model considers the water/steam pressure network for modeling via thermodynamic principles in cooperation with some other correlation equations in the form of polynomials for capturing the compound turbines' behavior. This approach of modeling is regarded to be hybrid between both empirical and physical methods to gain the physical insight as well as sufficiently accurate results. The work has been extended by the configurations of temperature and feedwater controls.

C. Wang et al. [43] have established a model for 660 MW for control and thermodynamic analysis of the plant. The model has been implemented in GSE software and the model parameters, which are based on thermodynamic properties of the fluid, have been packaged inside the GSE tool with the main fundamentals conservation equations. The model is distinguished also in simulating the transient processes and there has been considerable attention given to the startup and shutdown as well as the energy storage and control systems. However, only the results of load variations mode from 50% to 100% of the turbine heat acceptance have been reported.

Fan et al. [44] have built a dynamic model for 1000 MW coal-fired ultra-SCPP for the once-through mode. The model structure is based on mass and energy conservation principles and the model unknown parameters are regarded generally to be lumped parameters and identified by an improved version of intelligent optimization, which is an immune genetic algorithm (IGA) technique. For operating ranges of loading 600–872 MW and 954–880 MW, the model variation trends have been in promising agreement with the measurements for sufficient time of operation. By attaining appropriate accuracy of the model, the model validity and its suitability for control system synthesis has been proved.

Other models are built by similar principles of physics and parameters computation with emphasis on different objectives, such as evaluating the feasibility of $NO_x$ emissions reduction by the feed-water flow temperature [45], fluid heating in combustion chamber water-walls [46,47], studying energy storage strategies in SCPP [48], and fluidized bed combustion by hybrid model [49]. To sum up, the first principles of modeling has gained a leading position in modeling SCPP for many reasons, including proper accuracy for prediction of normal and emergency conditions, online monitoring,

and control system applications. The next subsection reviews the empirical method of modeling and highlights their aspects from control point of views.

### 3.2. Empirical Modeling of SC and USC Power Plants

Empirical models are built from operating data without derivation of the physical equations that govern the system. Such models are sometimes referred to as "black-box" models that eliminate the necessity for the knowledge of the internal behavior of the system. Zhong-xu et al. [50] have published a combined experimental/mechanism analysis model, with some nonlinear polynomial functions, and that has been fitted to 660 MW supercritical unit. Another more advanced interesting example for empirical models is the artificial neural networks (ANN) that has been presented in many published research articles with different architectures. Intelligent techniques, ANN in particular, have attained very accurate results for modeling supercritical plants.

K. Y. Lee et al. [51] have presented the approach of diagonal recurrent neural network (DRNN), which is extensively trained with filtered data trends gathered from plant simulator of 500 MW SCPP operating unit. The responses of many components have been easily captured by the ANN, which includes the flue gas network, the water/steam network, the coal mill or pulverizer, the turbine, and the generator units. For load change from 250 MW to the rated output (500 MW), the ANN have shown promising performance in simulating the plant variations with very accurate results. Other variables, which are the pressure and temperature have been also depicted. The developed combined model of ANN has been also compared against Rankine-cycle simulator to show the temperature-enthalpy relationship by both tools and again, the results have been satisfactory for three different rates of maximum generation.

X. J. Liu et al. [52] have modelled the behavior of 1000 MW ultra-supercritical power plant using two structures of ANN. One is based on ANN with radial bias function (RBF) and back-propagation learning algorithm, and the other is based upon neural fuzzy structure. Moreover, the method is compared with recursive least squares method (RLSs) for identification of autoregressive-moving average with exogenous variables (ARMAX) to show the effectiveness the ANN. The large load variations between 1000 MW and 600 MW have shown the effectiveness of the reported method for simulation of ultra-supercritical power plants when compared with real plant measurements. Notwithstanding, a numeric comparison of the root mean squared error (RMSE) has been reported to confirm the superiority of neural fuzzy method over the ANN-RBF and the ARMAX identified by RLS methods.

Zhang et al. [53] have published a more sophisticated structure of neural network, which is deep neural network (DNN) framework with stacked auto-encoder modeling approach for 1000 USC unit. In the training phase, the maximum correntropy has been selected to be the objective function because it relieves the abrupt changes that exist in the plant data. Consequently, very accurate results have been attained, which has brought the multi-layer perception (MLP), that has been adopted in [52]. The inspection of the RMSE in both approaches again indicates the superiority of the proposed technique.

Cui et al. [54] have constituted DNN for modeling and control of 1000 MW USC unit. Deep-belief network (DBN) has been used to capture the process of 1000 MW USC plant. The motivation behind the work has been to control the pressure, enthalpy, and the generated power of the USC unit process via economical model predictive control. In the control review section, the control part will be investigated for the paper. The DBN model has attained wide-range accurate results as the load varied between 525 MW up to 950 MW in the validation phase with very small RMSE. Although the neural fuzzy proposed in [52] has been more accurate in terms of the values of RMSE, the conventional ANNs experience complex computations when dealing with enormous data like what is normally gathered for USC power plants and DNN has proved its superiority for handling big data with ease. Thereby, deep-belief network has been established with 7 hidden layers to contain the 1000 MW USC process and the RMSE have been compared with multi-linear state-space models identified by subspace algorithm.

To sum up, some comparative points between physical and neural network approaches can be brought out. It might be seen that the NN approach is generally more accurate than physical models and wider in capturing more processes within the plant, especially for the generated power responses. Nevertheless, the NNs are pointedly dependent on the data-sets that have been used for training, which are mostly collected during normal load following mode and they probably fail to simulate some abnormalities or emergency conditions. For subcritical units, ANN has degraded to simulate the water-level control failure [21]. Moreover, for frequency simulations studies of subcritical combined cycle units, ANNs are not found to be preferable because they naturally round-up and down the very small frequency variations of untrained set of data with the values from the trained one, whereas the subspace state-space identification and simplified physical models have rather shown better frequency excursions simulation results [55]. As a consequence, the situation can be applicable in case of frequency responses and mechanical abnormalities of SC and USC units. It is then very hard, and even unnecessary, to decide which approach is ultimately better and the distinctive choice can be decided only according to the specific application. Table 1 summarizes the modeling survey for coverage from 1958 to 2020.

**Table 1.** Modeling review summary (coverage 1958–2020).

| Physical Modeling | | | Empirical Modeling | | Hybrid Models |
|---|---|---|---|---|---|
| **Parameters Computation** | **Processes Covered** | **Simulator Tool** | **Artificial Neural Networks (ANNs)** | **Algebraic Polynomial** | [42,49] |
| Thermodynamic or Thermo-Physical properties or formulas. [5,31,34,40–43,48] Thermodynamic properties (online computation) [46,47] Direct comparison with another detailed simulator or on-site measurements data [32,34–36] Multi-Objective Intelligent Optimization to fit with on-site measurements GA [3–5,37–39] IGA [44] Remark: Two references cited repeatedly in two approaches means that both methods are used so the model contains fixed and dynamic parameters. | Coal mill- Boiler turbine-generator [3–5,37,38] Coal mill-boiler-turbine [39,44] Boiler-turbine-Generator [45,48] Boiler-turbine [32,33,36,42,43] Boiler [34,35,40,41,46,47] Functional process integrated with CCP: [5,42] Operational processes: Startup and load following [40,43] Load following [3–5,31–48]. | APROS® [40,41] APD® [42] FORTRAN 95, SimuEngine [4] GSE [43] MATLAB® [3–5,37–39] Thermolib [3]. Remark: No detailed information is given in other references about the description of the computer tool. | Diagonal Recurrent Network (DRN) [51] Feed-forward Back-propagation(BP) with RBF [52] Deep Neural Network (DNN) [53,54] | [42,50] | |

## 4. Control Strategies of SC and USC Power

Thermal power plants, subcritical or supercritical, have drastically evolved in the past decades because of the advancement in the field of dynamic simulation and control engineering [56–97]. Optimal and predictive control theories have offered dominant approaches for many years ago and are expected to be the forthcoming control strategies with more complementary features for computing and time of the control action, which leads to process optimization at the supervisory level [91,92]. Nevertheless, the control of SC and USC units is far more complex than the control of subcritical units. In addition, the operational objectives of SCPPs are more sophisticated, which demands advanced control techniques.

Figure 7 summarizes the most salient modern control architectures and objectives. The objectives have been classified to general and specific, where the general requirements must be fulfilled for continuous operation and the specific requirements are subsequent requirements that provide corresponding advantage of cleaner and faster production to compete other generation technologies.

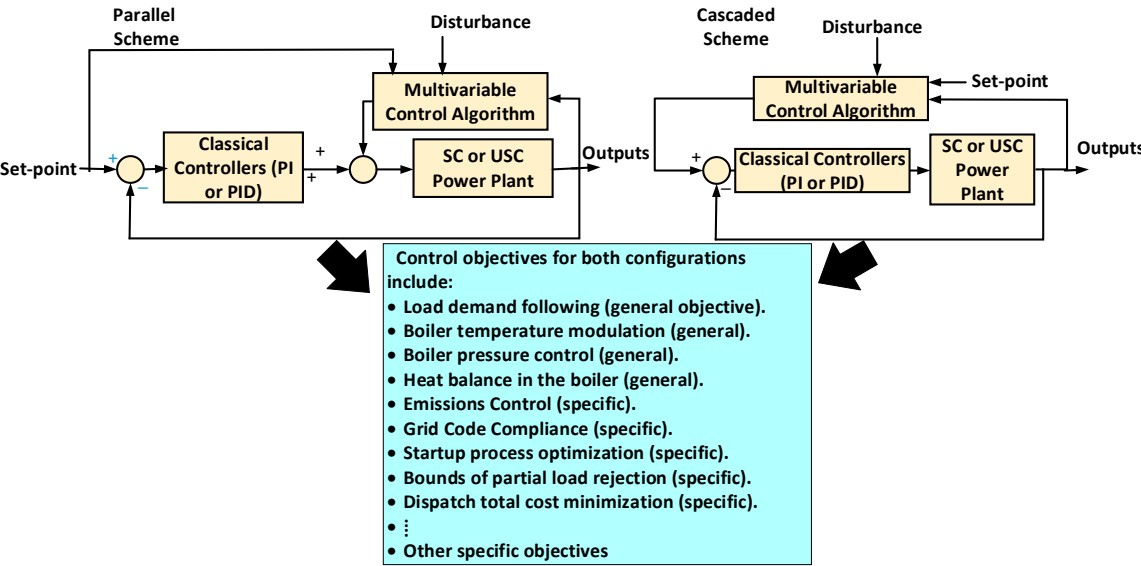

**Figure 7.** Modern control architectures and objectives.

H. Nakamura et al. [56,57] have gained the precedence for the first published optimal control of SCPP, which is implemented practically on a 500 MW SC oil-firing plant in 1978 [56,57]. The work has been offered to demonstrate the advantage of the offered optimal controller over the traditional PID controller. An autoregressive (AR) model has been identified to depict the behavior of the 500 MW SC units for digital controller development. Dynamic programming DP has been used to compute the gain matrix parameters of the state feedback controller and to minimize certain performance criterion. The controller has been implemented in a way to edit the signals of the conventional controllers for optimal control performance. It has been proved, from industrial and theoretical viewpoints, that it has been preferable to leave the conventional and apply the multivariable controller as supervisory layer over that regulatory layer. PID is considered in the system for two main reasons. First, the system identification procedure is done on the closed loop system, which is well-known as closed loop identification. Second, the existence of the classical controllers allows more flexible operation for the operators who have built their practical experience with the conventional controller.

However, more advanced control architectures have been introduced, which are hierarchical in their structures and feasible for industrial applications, such as model predictive control (MPC) applications [58–60]. Model predictive control outperforms the linear quadratic regulator in two points, first is that the MPC includes the practical constraints and the operational restrictions in the optimization problem, second the MPC algorithm applies the receding horizon principle and predictive strategy as shown in Figure 8. The predicted output shall follow the reference trajectory by optimal control moves. The control moves that make should be optimally calculated for a specific control horizon. Then, the control inputs or moves are implemented for the first sample and neglect the rest of the trajectory of the control horizon in order have the computation opportunity to repeat the whole process of prediction and optimization. The principle states that although the future optimal control signals are depicted, the real trajectory takes only the first sample and neglects the rest of the control signal. The processes of predicting the output, computing the errors between reference and predicted output, and computing the control signal changes, are repeated in every sample time. It is naturally uses the receding horizon principle and the optimization window is moved forward every sample time, which lead to suitability of online implementation. Henceforth, it can be used to improve the performance of SC and USC power plants.

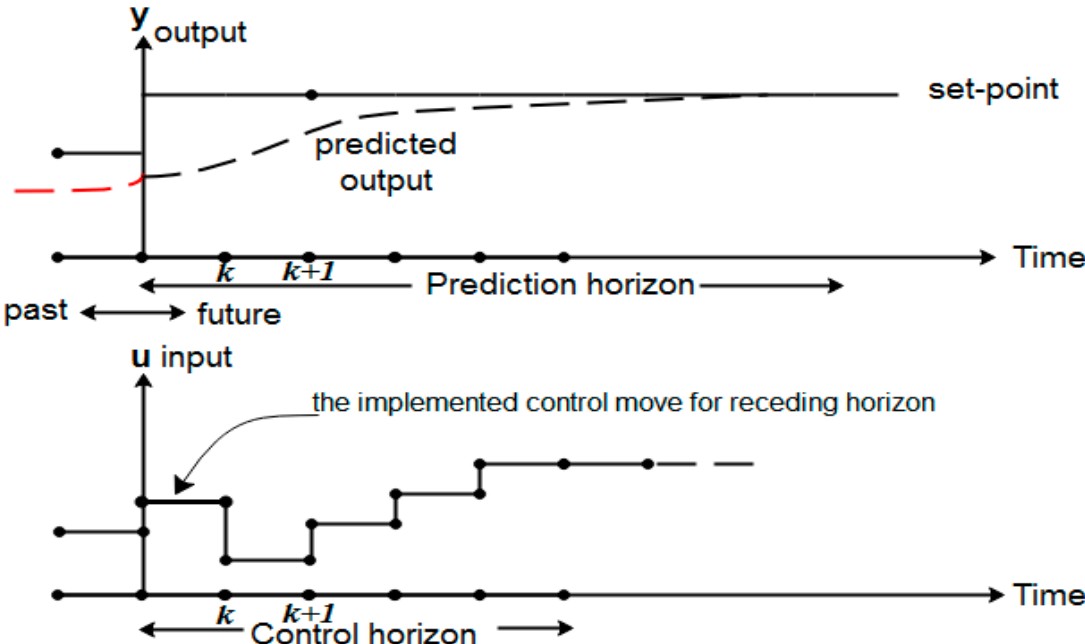

**Figure 8.** The basic idea of receding horizon control.

The literature provides many MPC algorithms, linear and nonlinear, which are either based on discrete-time mathematical optimization or intelligent optimization techniques. P. Gibbs et al. [61] have depicted introductory results of a workable, nonlinear, model based predictive control (NMPC), which allows a wider operating range than conventional MPC that uses linear models for prediction. The model used for predictions here is a simplified reduced order, first principle, nonlinear model. There is an unavoidable mismatch between the plant and model parameters if lumped parameters are used. To resolve this issue, the model parameters are identified online by nonlinear least squares and Kalman filter is used to estimate the plant states in real time. Then, the estimated states are used to calculate the future control inputs. The controller has shown effective results by matching the desired power signal for a gradual change of the real power by ± 30 MW around the current value, yet, the computation effort is obviously high and more practical solutions seem to be potentially discussed. It is more topical to focus on the practical objectives of the published controllers more than the control theory requirements.

A. J. Rovnak et al. [62] have offered multivariable controller based on dynamic matrix control (DMC) algorithm and integrated the controller to ninth order nonlinear SC plant model via simulation tool kit. A single matrix control model is utilized in the control algorithm to predict the plant output and compute the optimal control moves. The dynamic responses of all output controllable variables are included in the matrix model, which facilitates the optimal control strategy and outperforms other computationally demanding methods, such as algorithms that require Kalman Filter. The aforementioned three-inputs three-outputs modeling consideration is adopted in the paper and controller have shown promising performance for normalized load increase from 0.5 pu to 0.8 increase and adverse to 0.5 pu in around only 22 min real time simulation. However, the application is done on physical simulator kit rather than practical SCPP.

B. W. Shinohara et al. [35] have designed a nonlinear control system for SCPP which is inspired by the geometric structure of a simplified and physics-oriented phenomenological model. The intuitive control system have demonstrated its feasibility through power response from 515 MW to 450 MW. The controller has an advanced mathematical analysis and the application by simulation study has fairly shown the controller eligibility. However, from the concluded limitations of the proposed controller in that work, it has been concluded that the controller performance has been inadequate for steam pressure regulation.

Mohamed et al. [63,64] have applied linear model predictive control strategy on 600 MW SCPP. The control system philosophy is to speed-up the coal mill process by using the MPC raw coal signal to correct the reference of the coal mill local controller. Thereby, the coal mill exhibits higher kinetic energy for faster coal powder discharging in the furnace, which have lead to overall improvement in the dynamic response of the plant [63,64]. These contributions are extensions of authors' detailed modeling task published in [3,37,38]. Because the application has been made on detailed non-linear process model, some compensatory elements are included in the MPC algorithm. In [63], multiple disturbances are considered in the internal model in similar manner of using multiple internal models of MPC, but only the disturbance term is changed according to the unit load demand. Simulations of the proposed strategy have shown improved results in comparison to traditional milling performance for power reduction from 620 MW to 480 MW, then sudden increase to 640 MW in 160 min. In [64], generalized internal model structure is assumed where the system noises are added to the constant disturbances' term. The paper focuses on studying the effects of the coal mill dynamics on the overall power plant responses, and thus load variations in 160 min from 600 MW to 620 MW, then to 600 MW have clearly shown the positive influences of enhanced coal mill control on the dynamic response of the plant. The enhanced coal mill capability is found to be an extremely important factor that helps in coal combustion at the earliest possible time and hence faster power delivery to the grid.

Draganescu et al. [65] have presented DMC strategy for temperature regulation of 600 MW SCPP. The contribution of the work is to consider the malfunctioning in the milling process as a disturbance and test the controller performance in such abnormality. The process model has been nonlinear, whereas the prediction model in the predictive controller algorithm has been a controlled auto-regressive moving average (CARMA) model. The coal mill abnormality has been implicated as sudden stop in one of the coal mills and the performance are found to be more stable than traditional controllers because of the predictive nature of the DMC strategy.

T. Lee et al. [66] have designed a supplementary control system, based on an adaptive DMC, which modulates air/fuel ratio in order to reduce the undesired emissions form 1000 MW USC power plant. The controller regulates the reference signals of the fuel flow by the primary air fan to the pulverizer and the forced draft fan to the furnace. The reduction in NOx emission has been noticed in the DMC control case by the reduced air/fuel ratio during the time of transient changes in the simulation window. The application appears to be made on physical nonlinear model. The controller has shown effectives also on subcritical drum unit. However, intelligent techniques are expected to have superior performance for larger load demand variations. In the past two decades, intelligent controls have shown duel-advantage of capturing wider operation range and more process variables with ease [50–54,66,93].

Zhong-xu et al. [50] have published a generalized intelligent coordinate control system for supercritical unit. The control strategy is mixed between classical and fuzzy predictive intelligent control. The controller has been validated over a wide range simulations of load changes with different rates. Nevertheless, the control has been difficult to follow due to its complex structure.

K. Y. Lee et al. [67] have designed neural network-based modified predictive optimal control (MPOC) for 500 MW coal-fired supercritical power plant. In the control algorithm, the unit load demand is used to calculate the feedforward controller actions, then those feedforward controls are used as initial candidate control actions in an on-line identification system, the on-line system identifies the corresponding outputs and fed them to the MPOC. The new optimal control actions are computed in the MPOC via heuristic optimization technique, which is particle swarm optimization. Wide-range processes in the SCPP have been handled and included in the global control system of the plant, which are the milling process model, the air/gas path model, the water/steam process model, and the turbine-generator model. The controller have shown promising performance for power demand tracking from around 320 MW to 500 MW and modulating the steam pressure in subcritical and supercritical regions.

K.Y. Lee et al. [93] have extended the application of intelligent optimal control on an ultra-supercritical unit where reference governor is used to supply the system with feedforward control actions, which resolves the issue of intensive computations in the scheme published in [67]. The actual optimized control actions are provided by feedback control with intelligent gain tuning. The system have shown superior performance of long-range power demand control that has variations from 1000 MW sudden rejection to 600 MW, then sudden load application to 800 MW in 200 min.

Cui et al. [54] have developed economic model predictive control (EMPC) based on DNN to control the load demand of 1000 MW USC unit. Augmented model has been used to compensate the time delay in fuel preparation stage and nonlinearity in the plant. The controller performance has been guaranteed by closed loop stability analysis and through observation of load changes from 800 MW to 1000 MW and power decrease from 1000 MW to 650 MW.

From the survey investigation, it is readily deduced that MPC control technique is the most salient methodology for control system development of fossil fuel SC units regardless of whether it is based on artificial intelligence applications or based on discrete mathematical analysis and mathematical optimization. Table 2 summarizes the configuration for every control strategy published, which ensures the eligibility of both schemes.

**Table 2.** Control configuration summary (coverage 1978-2020).

| Parallel Scheme of Multivariable Control for SC and USC Power Plants' Automation. | Cascaded Scheme of Multivariable Control for SC and USC Power Plants' Automation |
|:---:|:---:|
| [50,54,56,57,61,62,67,93] | [63–66] |

## 5. Proposed New Trends

SC and USC operation strategies of power plants have brought cleaner fossil-fuelled units, which occupy a primary choice for future conventional generation technologies. Therefore, it is expected that the research attempts will continue to develop more accurate models and more advanced control strategies for more efficient and1 cleaner operation of this generation technology. Throughout the wide-range survey, there are some obvious opportunities for future researchers and postgraduate students, which can be suggested as follows:

1. Some system identification methods are found to be rewarding and have not been applied and verified on SC or USC power plants. Among the strong candidates of models, it has been found that Wiener, Hammerstein-Wiener, enhanced Wiener models, and Hammerstein structures, all worth investigation as they have brought notable improvements on modeling of other energy sources [98,99]. The approaches can be readily applied on supercritical and ultra-supercritical units and compared against other techniques.

2. For physical and mathematical model part, there are some other parameter identification techniques that are found to be more robust than genetic algorithms and other evolutionary computation techniques. The proposed method in this section is to use gravitational search algorithm (GSA), which outperforms the evolutionary computation techniques in parameter identification of another energy conversion system [100]. Therefore, it is believed that the GSA approach will introduce enhanced results for SCPP and USC models' identification.

3. There is a lack of intelligent control applications on physics-based models as most reported articles for intelligent control of SCPP have been applied on neural network based models. Physics-based models are more compatible with system physics under normal and emergency conditions and offer the advantage of being easy to observe/estimate the intermediate variables, which are very difficult to observe or measure in practice. On the other hand, the NN controller is found to be more rigorous in following large load applications or rejections. Studying SCPP by this combination seems to be attractive and likely will bring improvements in the dynamic responses of SC and USC units.

4.  Startup process optimization is not extensively studied and seems to be forthcoming research proposal. Although these are published articles in this context [40,43], there are many alternative methods that can be applied. Multiple model predictive control (MMPC) is expected to a leading choice to optimize the startup process of SCPPs. Identified linear state-space models are preferable to cover the entire range of startup process in the MMPC algorithm and to facilitate the computation demands. Then, a comparative study can be conducted with and without MMPC to confirm how the proposed technique helps the operators in integrating the plant to the gird in a shorter time.

## 6. Conclusions

In this paper, a survey of the research reported and state-of-the-art techniques for modeling, simulation, and control of supercritical and ultra-supercritical fossil-fired power plants is presented. The main importance of this review paper is to easily clarify the way for the future research in this area. The main findings of the review are explained point-by-point as follows:

- The modeling review part has been classified into physical models and empirical models, in which the physical models includes the mathematical models that rooted originally and structurally from system physical and engineering principles, whereas the empirical models include data-based models that are built in the computer algorithm without any physical derivations. Although the later is found to be generally more accurate over wide range of operation, the former approach has the necessary physical verity that makes the intermediate variables that are difficult or expensive to measure available by reasonable inference. Furthermore, the simulations during emergency conditions, such as sudden loss of one of the mills, failure in the water-level control, and frequency excursions have given priority to physical models. Therefore, both methods have doubtless eligibility in modeling SC and USC units because of the salient merits of each method. The future recommendations for this part is to apply alternative system identification techniques as those proposed in Section 5 for more adequately accurate results. The proposed alternative model structures and identification algorithms have given prominent enhancement for other energy sources so they can be promising also in case of SC and USC processes.

- For control part, the review has focused on the control performance in load following capability, computation burdens, and the control scheme or configuration. Other modern objectives are included in recent control strategies, such as emissions control, energy efficiency improvements, and milling performance effects. The review reports the main articles, each has either parallel or cascade structure of two control levels: supervisory and regulatory. Despite all these promising achievements, there are still many issues to consider in the future as the ordinary way of control system philosophy is no longer sufficient to accomplish with all contemporary requirements. Intelligent techniques are one of the leading options, which need considerable background inartificial intelligence and power generation plants. The literature is also lack of extensive study for startup process optimization, which can be done by MPC, explicitly MPC or MMPC.

- The future recommendation is to study the suggested new points that are presented as proposed new trends in Section 5. Some state-of-the-art intelligent techniques for parameter identification and control applications are suggested that have not been applied to SCPP yet. It is believed that, these concise proposal open the way for more intensive research proposals in this area for future researchers subject to having favorable facilities, attaining the appropriate standard of the research, and hence more improved results.

**Author Contributions:** O.M. reviewed the literature, developed the survey, and wrote the manuscript. A.K. revised and improved the paper. J.W. initiated the work topic and supervised the work during PhD studies. All authors have read and agreed to the published version of the manuscript.

**Funding:** This research received no external funding.

**Conflicts of Interest:** The authors declare no conflict of interest.

## Abbreviations

| | |
|---|---|
| APD® | Aspen Plus Dynamics® |
| APROS® | Advanced Process Simulation Software® |
| ANN | Artificial Neural Networks |
| ARMAX | Autoregressive-Moving Average with Exogenous |
| CCGT | Combined Cycle Gas-Turbine |
| CCP | Carbon Capture Process |
| DMC | Dynamic Matrix Control |
| DNN | Deep Neural Network |
| DRNN | Diagonal Re-current Neural Network |
| ECON | Economizer |
| GA | Genetic algorithm |
| GB | Great Britain |
| HP | High pressure |
| HRSG | Heat Recovery Steam Generator |
| IP | Intermediate pressure |
| MPC | Model Predictive Control |
| MMPC | Multiple-Model Predictive Control |
| NMPC | Nonlinear Model Predictive Control |
| NN | Neural Network |
| RH | Reheater |
| SC | Supercritical |
| SCPP | Supercritical Power Plant |
| SH | Superheater |
| USC | Ultra-Supercritical |

## Symbols

| | |
|---|---|
| $E$ | Energy or Energies (KJ) |
| $K$ | Unknown Fixed Parameter |
| $m$ | mass (Kg) |
| Q | Heat Transfer (KJ) |
| t | Time (seconds) |
| u | Input |
| x | State variable (pressure or temperature for a heat exchanger) |
| y | Output |

## Subscripts

| | |
|---|---|
| *in* | input |
| *out* | output |
| *rh* | Reheater |
| econ | Economizer |
| *sh* | Superheater |
| *ww* | Waterwall |

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
