# Peer review of "Modeling and Control of Supercritical and Ultra-Supercritical Power Plants: A Review"

_energies, doi:10.3390/en13112935_

Round 1
Reviewer 1 Report
>>>Content issues:
Some info addressing very old references (e.g. [31]....[34]) should be removed from the paper or at most be reduced.
Sometimes useless numerical info are provided (e.g. the powers addressed by a paper)
>>>English issues and typos:
Suggestion to rephrase the expression starting with the line 146”: „are : (a) the uniform properties of fluid; (b) the steady flow process; (c) excluding the flue gas path and so on.” (or something similar, not compulsory with (a), (b)...)
Line 149: „into components, each component” -> „into components. Each component”
Rephrase the statement starting at line 155.
Line 158: „there is will be a need” – needs revision
Line 314 „stat?-space”
Line 315 „with load varies????”
Line 481 „advantage of for cleaner???”
Line 496: „reasons: Firstly”
Line 616: „Tabe.2”
Line 647: „very difficult to observer?? or even measure”
Line 658: „as soonest??? as possible”
Line 173 -> „it will be judicious to report” perhaps should be substituted by „one should present”
Line 255 -> a verb is missing.
Phrases starting at line 108 ,371, 470,476, (500 to 511), 570, 590 need revision.
>>>Format
Extra space at line 140
Fig. 3 -> instead „..” use „ ...”
In Figs. 3 and 4 some symbols are truncated on the right edge. Fig. 5 (on page 8) – truncations or incomplete representation of some symbols.
Fig. caption no. 4 is duplicated.
Extraspaces at the end of certain pages need to be removed (e.g. page no. 7, 13)
Fig. 6 is distorted (longer on vertical axis)
At the beginning of page no. 15, „3.” Is the number of a new section ? If yes (less probably) , it should be „4.” and the sections numbering needs revision. Otherwise , use 3.3.
Sometimes „et all” is written with italic fonts , sometimes with regular fonts.
Reviewer 2 Report
In this study, a comprehensive review of the process modelling of supercritical power plants and their related controls was presented. The review has a good structure and the overall technical and scientific quality is fair. The use of English is satisfactory and the article can be followed easily. The title accurately reflects the study. The objective is well defined and I have no criticisms regarding the interpretation of results of reviewed papers. The questions are as follows:
- In the literature, I found some similar reviews. So the authors are asked to show their original contribution regarding the objective of this review in a more convincing way.
- The introduction should be rewritten with a more careful explanation of the main findings of this review.
- To make the conclusion section more clear, authors are highly encouraged to include the point-by-point findings of this article. The current conclusion is written very wide and it is not easy to maintain the key findings.
- The quality of the Figures and their legends are not sufficient. Furthermore, I found difficult to read the text in some Figures. Please try to enhance Figure's resolution.
Conclusion: In my opinion, the topic of this review of relevance. However, authors are asked to improve the paper considering the above comments to recommend this paper for publication in Energies. My Final decision: revise.
Reviewer 3 Report
Congratulations on an interesting idea for a patent and article.
The team of authors reviewed specialist literature related to the topic of the article. The conclusions drawn are interesting for researchers of this issue.
There are small errors in this article that should be corrected:
- It is necessary to make small corrections in the article, certainly the quantities appearing in equations from 1 to 7 should be explained.
- Figure 7 requires a description of the units on the time axis and the size (y) - input and output.
I have little attention to the authors, the conclusions lack a description of directions for further work and research.
Reviewer 4 Report
This paper presents a Review of the Modeling and Control of Supercritical and Ultra-Supercritical Power Plants.
This study aims to review the literature and analyze the different control methods used in this type of plants for a long period of time.
The manuscript is well organized and easy to follow. It provides an extensive and interesting analysis of the development of modeling to control this type of plants.
The objectives are very clear in the introduction section. But the conclusions are not very clear.
I suggest a major revision of the manuscript for that it can be accepted.
The remarks are:
- The references are not current in their majority.
- Define abbreviations and nomenclature to be placed on the first page of the article.
- Review English. Among others:
. Convecction, lin.102
. Lin 470 dramatically? Drastically
. Lin. 658 gird? Grid
. Lin. 673 excursions?
- Clarify between lines 92 and 93.
- The conclusions should be more clarifying.
You check numbering sections.
Reviewer 5 Report
This is very interesting review. In Introduction Authors adduced on a number of papers devoted to the research of the modelling and control of supercritical and ultra-supercritical power plants. In the paper, the research reported and state-of-the-art techniques for modelling, simulation, and control of supercritical and ultra-supercritical power plants was showed. In paragraph named “Coal-Fired Supercritical Power Generation Process” authors describe the energy conversion process of SCPP. Next, the supercritical power plant modelling for dynamic performance studies were showed. The authors take into consideration two categories: physical and mathematical modelling of SC and USC power plants and empirical modelling of SC and USC power plants. In “Control Strategies of SC and USC power” paragraph the optimal and predictive control theories was presented. At the end of paper the new trends of SC and USC operation strategies of power plants was showed.
I have a one comment:
- It is quite strange that in Figure 2 from the slag hopper, ash is fed into the chimney. Flue-gas should be led to the chimney from the boiler outlet. Incorrect designation in the diagram.
Round 2
Reviewer 2 Report
I would like to thank the authors since they have precisely reviewed their article and answered sufficiently all comments. I don’t have any further comments. The paper can be accepted for publication as it stands.
Reviewer 4 Report
The authors have successfully responded to comments requested, thereby improving the content of the manuscript. In this state, the manuscript has been recommended for publication.